# Whole-Body Vibration (WBV) as a Conditioning Activity for Roundhouse Kick (*mawashi geri*) Performance in Karate

**DOI:** 10.3390/jfmk9030145

**Published:** 2024-08-26

**Authors:** Johan Robalino, Lucieli Teresa Cambri, André Cavalcante, Emerson Franchini, Bruno Mezêncio, Jacielle Carolina Ferreira

**Affiliations:** 1Postgraduate Program in Physical Education, Federal University of Mato Grosso, Cuiabá 78060-900, Brazil; johanrobalino@gmail.com (J.R.); andre_cavalcante_santos@hotmail.com (A.C.); 2Martial Arts and Combat Sports Research Group, Sport Department, School of Physical Education and Sport, University of São Paulo, São Paulo 05508-060, Brazil; efranchini@usp.br; 3Biomechanics Laboratory, School of Physical Education and Sport, University of São Paulo, São Paulo 05508-060, Brazil; mezencio@usp.br

**Keywords:** post-activation performance enhancement, impact force, speed, kinetics, kinematics

## Abstract

Karate athletes strategically use lower-limb techniques in combat, with the roundhouse kick (*mawashi geri*) being highly effective in kumite. To quickly improve the technical performance before training or competitions, conditioning activities (CAs) are often utilized. Recently, Whole-Body Vibration (WBV) has emerged as a potential conditioning activity (CA). This study aimed to analyze the acute effects of WBV as a CA on the performance of the *mawashi geri*. The sample included sixteen male karate athletes. The study had a familiarization and two experimental sessions: one with WBV and the other without (NWBV), conducted randomly and counterbalanced, each preceded and followed by a *mawashi geri* assessment on a force platform. During the CA intervention, the participants performed four sets of isometric half-squats on a vibration platform at a frequency of 26 Hz and an amplitude of 4 mm in the WBV condition, while the platform was off in the NWBV condition. A significant reduction in the *mawashi geri* attack phase time was observed under the WBV condition [pre: 0.31 ± 0.03 s; post: 0.30 ± 0.03 s] compared to the NWBV condition [pre: 0.31 ± 0.04 s; post: 0.32 ± 0.03 s] (*p* = 0.02). However, no differences were noted regarding the impact force or other kinetic variables between the conditions. Therefore, WBV did not increase the performance of the kinetic and kinematic variables of the *mawashi geri* in karate athletes, but it is possible that there is a positive effect on attack time, suggesting that further studies with different vibration protocol configurations would be beneficial.

## 1. Introduction

Karate-Do has undergone a remarkable evolution, transitioning from a traditional martial art to a globally recognized sport [1]. In competitions, two modalities predominate: kata and kumite [2]. Sport kumite is a controlled-contact match between two athletes, aiming to score points through high-speed and powerful techniques. The rules permit full-force application to the trunk while maintaining controlled contact to the head [3,4]. Tabben et al. [1] demonstrated that elite karate athletes who win tend to employ more lower-limb techniques, including kicks to the head and trunk, which are strategically advantageous due to their high scoring in kumite [5]. Among the lower-limb techniques, the roundhouse kick (*mawashi geri*) is considered the most used, effective, and potentially dangerous [6].

To acutely enhance the performance in technical actions such as the *mawashi geri* in training and competitions, a variety of stimuli, known as conditioning activities (CAs), are commonly used [7]. These stimuli have the potential to induce Post-Activation Performance Enhancement (PAPE) [8,9,10], resulting in acute increases in maximal strength, power, and speed [8]. The benefits of PAPE with different conditioning activities are well documented, and the improved performance is the result of enhanced muscle activation and neural drive [9,10], explained by underlying mechanisms that include increased myofilament sensitivity, enhanced neural recruitment, elevated body temperature, and muscle water content [9,11].

In combat sports, various forms of CA are explored. For example, the use of elastic resistance during kicks has been shown to significantly increase the kick speed and muscle activity in the vastus medialis and rectus femoris [12]. Additionally, plyometric exercises and repeated high-intensity techniques have been demonstrated to improve the values in the countermovement jump test and speed in the Frequency Speed of Kick Test (FSKT) in taekwondo athletes [13]. In the context of karate, a warm-up using CA with resistance bands, representing 10% of the individual’s body mass, resulted in significant improvements in the long jump values (2.2 ± 1.7%) and left punch reaction time (−7.3 ± 5.9%) [14].

Recently, Whole-Body Vibration (WBV) has emerged as a possible CA. WBV induces high gravitational loads through vibration frequency and amplitude that act as a preloading motion and can cause post-activation potentiation mainly downstream of the neuromuscular junction [15]. Although the only study found that tested the effect of WBV on the kicking technique did not find an increase in the roundhouse kick performance in taekwondo athletes [16], studies that examined the acute effect of vibration on other motor actions found improved performance in sprints [17], vertical jumps [18,19], and power [19,20,21]. The advantages of using WBV as a CA are mainly the ease of application and the low risk of musculoskeletal injury during exposure. However, the interaction of mechanical vibration with human tissue is a complex phenomenon, making it challenging to determine whether WBV as a CA would be able to promote enhanced performance for a complex task such as the *mawashi geri*.

The protocols with WBV applied to different populations have shown a potential effect on power and strength performance, especially in less specific tasks taking into account combat sports, such as vertical jumps and isometric strength tests [21,22,23,24]. Considering that the study evaluating WBV in a specific combat sports task utilized a single one-minute series of WBV as the CA [16], this exposure time is considerably shorter than those employed in other WBV studies [21,22,23,24]; it still remains uncertain whether mechanical vibration can be a potential AC to induce increased performance in a complex and specific task in combat sports. Therefore, the objective of this study was to analyze the acute effects of WBV on the *mawashi geri*, encompassing kinetic and kinematic variables in karate athletes. Our hypothesis is that WBV can potentially optimize the specific kinematic aspects of the *mawashi geri* performance related to an increase in impact force and a reduction in the execution time of the *mawashi geri*.

## 2. Materials and Methods

### 2.1. Participants

The minimum sample size for this study was calculated using G*Power software (version 3.1.9.4; Düsseldorf University, Düsseldorf, Germany). *T*-tests were employed to determine the power based on the study design, with the following parameters: difference between two dependent means, α error probability = 0.05, effect size = 0.90, and power (1−β error probability) = 0.80. This analysis indicated that the minimum sample size required for statistical significance, achieving a real power of 0.83%, was at least 10 participants. Consequently, the participant group in this study comprised sixteen male karate athletes, including two brown belts and fourteen black belts, all practicing *Shotokan* style at a national competition level (age = 25.6 ± 7.1 years; height = 1.74 ± 0.05 m; body mass = 71.5 ± 8.7 kg; body fat percentage = 14.7 ± 6.7%; experience of training = 11.0 ± 4.9 years).

The inclusion criteria required that participants had engaged in strength training and karate-specific training for at least one year (i.e., tactical, technical, and physical karate-specific training at least three times a week for a minimum of one year) and had competed in at least the most recent national championships. Additionally, participants were required to have no musculoskeletal injuries in the six months preceding the study, and all athletes during the data collection period were in a transitional phase as they had no scheduled competitions for the remainder of the year.

The participants were informed about the nature of this study and signed an informed consent form. All procedures were approved by the Research Ethics Committee of the Federal University of Mato Grosso (UFMT) under CAAE number 33016620.3.0000.8124.

### 2.2. Study Design

Data were collected across three separate visits, with a 48 h interval between them, coinciding with the athletes’ regular training periods. In the first session, subjects signed the informed consent form, underwent anthropometric assessment, and were familiarized with the evaluation method for the *mawashi geri*. In the second visit, block randomization was implemented. The first participant was randomly assigned to a condition, and the subsequent participant was assigned to the opposite condition, with this process continuing for all participants. The application of experimental conditions was randomized, with one condition involving Whole-Body Vibration (WBV) and another without WBV (NWBV). For the third visit, the experimental conditions were counterbalanced. Each intervention was preceded and followed by an evaluation of the *mawashi geri*, as illustrated in Figure 1. All study procedures were performed in a climate-controlled laboratory environment, with temperatures maintained between 20 and 24 °C.

### 2.3. Familiarization

During the first visit, participants were familiarized with the experimental procedures, completed and signed the informed consent form, and had their anthropometric data collected. For the anthropometric measurements, the following instruments were used: a Multilaser HC021 digital scale (Manaus, Brazil, precision of 0.1 kg), an Avanutri stadiometer (Três Rios, Brazil, precision of 0.1 cm) to measure height in the bipedal position, and a Harpenden C-136 analog caliper (Burgess Hill, UK, precision of 0.1 mm) to estimate the body fat percentage using the seven skinfolds protocol of Jackson and Pollock [24].

### 2.4. Conditioning Activity Intervention

Participants were exposed to two experimental CA conditions: the Whole-Body Vibration (WBV) condition and the Non-Whole-Body Vibration (NWBV) condition. Before applying each protocol, athletes performed the same standardized warm-up as in the familiarization session, which consisted of 10 min on a cycle ergometer followed by 3 submaximal repetitions of the *mawashi geri* kick. In both experimental conditions, participants executed four sets of one-minute isometric half-squats, maintaining a constant knee angle of 130°, with feet parallel and shoulder-width apart, on a side-alternating vibration platform (P240I KIKOS^®^, São Paulo, Brazil). The rest interval between sets was one minute. In the WBV protocol, participants were exposed to sinusoidal vibrations at a fixed vibration frequency of 26 Hz and 4 mm amplitude. Using a triaxial accelerometer at the base of the platform, fixed at a location close to where the feet are positioned, we found an acceleration of 3.78 g at 26 Hz. Previous studies by our study group using the same vibrating platform show the consistency of the platform measurements and the characterization of the stimulus at different frequencies [25,26]. In the NWBV condition, no vibration was applied (the vibration platform remained off). In both conditions, subjects remained barefoot while on the vibrating platform.

### 2.5. Mawashi Geri Test

This study was conducted in a laboratory setting, where the *mawashi geri* evaluation took place two minutes after the warm-up and two minutes after the application of CA (WBV or NWBV). The *mawashi geri* is classified as a swing-type kick, characterized by a sequence of hip flexion followed by rapid knee extension until impact with the instep [27]. The athletes performed the *mawashi geri* technique on a force measurement device, with intensity comparable to a real combat situation, using their preferred leg at a self-selected distance. The procedure began with the attacking foot positioned back, aligned with the reference tape on the contact mat. After a few seconds for concentration, the kick was executed according to karate specifications. To be considered valid, the attacking foot needed to impact the reference tape on the device and return to the initial position on the contact mat. Each attempt’s validity was assessed by a single evaluator, a 2nd dan black belt with 20 years of experience in karate practice. The test consisted of three valid attempts, with a one-minute interval between them. For the statistical analysis, the mean of the three attempts was used. 

The device used to evaluate the *mawashi geri* comprised four force transducers (S-type load cell DYLY 103, Calt, Shangai, China), a contact mat (Multisprint. 3.5.7, Hidrofit Ltd., Belo Horizonte, Brazil), a microcontroller (Atmega 328 microcontroller, Microchip Technology, Chandler, AZ, USA), and four amplifiers (AD620, Analog Devices, Wilmington, DE, USA). This integration is as illustrated in Figure 2 and allowed for a sampling frequency of 1 kHz. A previous study by Robalino et al. [28] assessed the reproducibility of the *mawashi geri* evaluation procedure in karate athletes. This study involved 24 participants (mean age: 28.0 ± 10.2 years; height: 1.74 ± 0.07 m; body mass: 77.1 ± 13.9 kg; body fat percentage: 17.1 ± 8.5%; experience of training: 12.1 ± 7.5 years) who performed three repetitions of a *mawashi geri* on two separate occasions. The reliability assessment yielded the following results: impact force (Pearson correlation coefficient, [r] = 0.96; intraclass correlation coefficient [ICC] = 0.93; coefficient of variation [CV] = 0.12%); impulse (r = 0.95; ICC = 0.93; CV = 1.08%); attack time (r = 0.82; ICC = 0.70; CV = 0.14%); contact time (r = 0.72; ICC = 0.60; CV = 0.18%); return time (r = 0.80; ICC = 0.71; CV = 4.28%); and kick time (r = 0.84; ICC = 0.77; CV = 2.88%).

The Matlab^®^ software R2017a was used for signal processing and analysis of the following dynamic variables: impact force (determined as the maximum perpendicular value on the device surface) and impulse (calculated as the integral of the force with respect to the contact time). The total execution time of the kick was divided into attack, contact, and return phases, measured from the foot losing contact with the contact mat on the ground until it touched the contact mat again. The kick variables are represented in Figure 3.

### 2.6. Statistical Analysis

The normality of data distribution was tested using the Shapiro–Wilk test, confirming that the values of all dependent variables conformed to a normal distribution. To evaluate the absolute and relative reliability measures between pre-test values of each day (WBV vs. NWBV), the standard error of measurement (SEM) categories was defined as follows: SEM < 5% (low), 5% ≤ SEM ≤ 10% (moderate), and SEM > 10% (high), and ICCs were calculated classified as follows: ICC < 0.5 (poor reliability), 0.5 ≤ ICC < 0.75 (moderate reliability), and ICC ≥ 0.75 (high reliability), respectively. Comparison of means between different conditions and moments for the dependent variables impact force, impulse, duration of attack, contact, return phases, and total kick time was conducted using a two-way repeated measures analysis of variance (ANOVA) (condition: with and without vibration; moment: pre and post). Effect size was estimated using the partial eta-squared (η^2^), with thresholds set at η^2^ = 0.01 for small effect, η^2^ = 0.06 for medium effect, and η^2^ = 0.14 for large effect. Mauchly’s test of sphericity was applied, and degrees of freedom were adjusted using the Greenhouse–Geisser correction when necessary. For the adjustment of confidence intervals, the Bonferroni correction was used. Statistical analysis was performed using IBM SPSS Statistics version 23^®^, with a significance level set at *p* ≤ 0.05.

## 3. Results

The pre-test data between the intervention days showed moderate to low SEM values for impact force (13.14%), impulse (9.97%), attack time (5.45%), return time (7.37%), total kick time (5.02%), and contact time (3.12%). In terms of the ICC, the values indicate moderate reliability for impact force (ICC = 0.71), impulse (ICC = 0.69), return time (ICC = 0.48), and total kick time (ICC = 0.54), whereas attack time (ICC = 0.43) and contact time (ICC = 0.42) exhibit poor reliability. Table 1 presents the mean and standard deviation values of the kinetic variables, as well as the results of the ANOVA. None of the kinetic variables analyzed exhibited a significant effect of condition, moment, or interaction, and the effect size values were low (impact force: condition η^2^ = 0.00, moment η^2^ = 0.00, interaction η^2^ = 0.00; impulse: condition η^2^ = 0.00, moment η^2^ = 0.02, interaction η^2^ = 0.00).

The kinematic results, as shown in Figure 4, revealed a statistically significant difference in the attack phase of the *mawashi geri* between condition WBV [pre: 0.31 ± 0.03 s; post: 0.30 ± 0.03 s] compared to NWBV [pre: 0.31 ± 0.04 s; post: 0.32 ± 0.03 s] (*p* = 0.02; η^2^ = 0.05). No significant differences were found in the contact phase (*p* = 0.14; η^2^ = 0.00), kick return time (*p* = 0.57; η^2^ = 0.00), and total kick time (*p* = 0.78; η^2^ = 0.00) between NWBV and WBV. There was no main effect for the moments (*p* > 0.05; η^2^ < 0.04), and, additionally, no significant interactions were identified between the conditions (NWBV vs. WBV) and moments (pre- vs. post-test) for any of the variables (*p* > 0.05; η^2^ < 0.03).

## 4. Discussion

This study aimed to investigate the acute effects of WBV on the kinetic and kinematic variables of the *mawashi geri* in karate athletes. The initial hypothesis suggested that the acute application of WBV as a CA would promote an increase in the impact force and reduce the execution time of the *mawashi geri*. However, the results obtained showed no significant effect of the condition or moment on the analyzed kinetic variables. Regarding the kinematic variables, there was a condition effect on the duration of the attack phase, with the athletes performing the attack phase in less time in the WBV condition. However, this result should be interpreted with caution as there was no interaction effect between the condition and moment. As the performance did not increase after the CA, the study hypothesis was refuted.

To the best of our knowledge, this study is pioneering in exploring the impact force of a kick technique after a CA with mechanical vibration application, making it difficult to compare with other studies. However, Třebický et al. [29] used a typical warm-up as a CA, composed of general and specific parts of combat sports (general warm-up, mobilization, and dynamic exercises specific to combat sports), without demonstrating any effect of the warm-up on the impact force of a rear punch. Similarly, in the present study, no differences were found in the kinetic variables (impact force and impulse) of the *mawashi geri* after a CA with WBV. It is noteworthy that there was also no main effect of moment, showing that, regardless of vibration, the isometric squat sets did not influence the force and impulse of the kick. A possible explanation for the lack of effects of the CA protocols on the kinetic variables of the kick could be that impact force depends not only on the force generated and the mass involved but also on the duration of contact and the execution technique [27,30]. Therefore, the complexity of the *mawashi geri* movement may be limiting the possible effects associated with the use of WBV as a CA [31].

Regarding the kinematic variables, our findings revealed a statistically significant reduction in the attack phase time for the experimental conditions. The attack phase is usually the most important during karate combat as it is the most offensive and evaluated by judges during kumite [32,33]. Oliveira et al. [16] investigated the acute effect of WBV on the roundhouse kick in taekwondo athletes, finding no significant differences in the attack phase time. Although the authors also used a frequency of 26Hz, in the present study, the number of sets and the recovery period were increased, resulting in a considerable increase in the vibration exposure volume, similar to the studies by Bosco et al. [34] and Kurt and Pekunlu [35], both carried out with athletes, who also applied approximately five minutes of exposure to vibration and observed an increase in the upper limb performance.

Mechanical vibration exerts an influence on the human body, manifesting through various physiological mechanisms [36], primarily through the activation of the neuromuscular spindle in response to rapid variations in the musculotendinous length. This mechanism triggers an active response to protect the internal structures, resulting in increased muscular activity and internal temperature [15,22]. This muscular activation and temperature increase may comprise the mechanisms supporting PAPE, which promotes a temporary increase in performance following a CA [8,9,10]. An advantage of using WBV over other CA protocols is the practicality and low risk of injury as it generally does not involve explosive actions.

Our findings did not show a moment effect for any of the variables investigated and under any of the experimental conditions; therefore, PAPE did not occur. However, the shorter duration of the attack phase in the WBV condition, even considering both moments, may suggest a possible potential effect to activate PAPE before training or competition in combat sports, particularly focused on improving the *mawashi geri* attack time. Moreover, it is essential to acknowledge the limitations of WBV as selecting appropriate protocols to achieve an efficient dose–response relationship remains challenging. Furthermore, our study’s sample consisted exclusively of male athletes practicing *shotokan* karate, which limits the generalizability of the results to other populations and karate styles. The individual variability among the participants could introduce variability in the results. Therefore, it is necessary to explore different types of protocols for well-trained populations, and WBV may not be specific enough regarding the motor skills required in certain sports, raising questions about the ecological validity of the results [37]. Furthermore, the effects of WBV on impact force have not yet been evidenced. We suggest that additional stimuli may be necessary, which could be reflected in more vibration sets or combining vibration with other CAs.

This study contributes to the understanding of how WBV as a CA can enhance the speed of execution of the *mawashi geri*, although the variability in effects and the applicability of these findings in broader contexts must be considered. Coaches may consider incorporating WBV into warm-up routines to optimize the performance in specific training sessions and even in competition. Further research should focus on tailoring WBV protocols to individual needs and exploring the impact on other karate techniques.

## 5. Conclusions

Based on our findings, our study demonstrated that the application of WBV as a CA did not affect the kinetic variables, such as impact force and impulse, of the *mawashi geri* in national-level karate athletes. However, a significant improvement in the attack phase time of the *mawashi geri* was observed under the WBV condition, but without the effect of moment. These results show that there was no PAPE under any condition, but they may suggest a potential benefit of WBV in accelerating the offensive phase of the strike.

Despite these promising results, the lack of effects on the kinetic variables and the individual variability in response among the participants highlight the need for further exploration of the WBV protocols. Specifically, it is necessary to determine the optimal dosage and vibration parameters to develop tailored protocols. Future studies could investigate whether, with appropriate adjustments in the duration and frequency of WBV use, the technical performance in combat sports and potentially other sporting disciplines can be improved.

## Figures and Tables

**Figure 1 jfmk-09-00145-f001:**
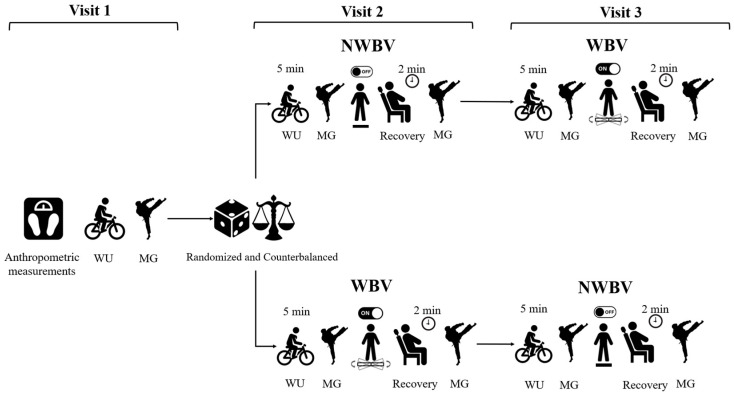
Experimental procedures. WU: warm-up; MG: *mawashi geri*; NWBV: No Whole-Body Vibration; WBV: Whole-Body Vibration.

**Figure 2 jfmk-09-00145-f002:**
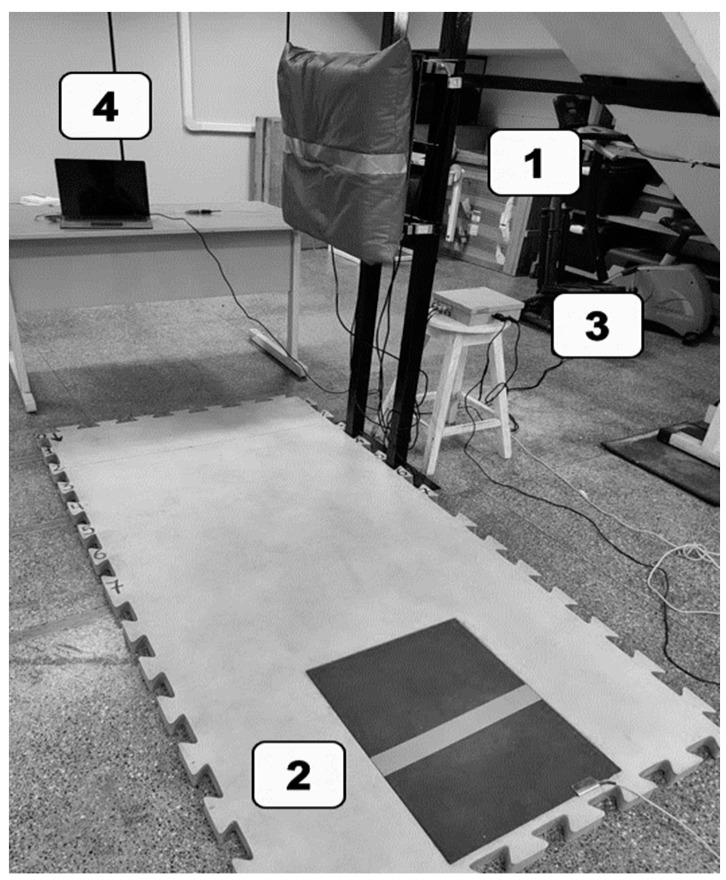
System configuration. 1. Load cell platform; 2. Jumptest^®^ Contact Mat; 3. microcontroller and amplifiers; 4. computer.

**Figure 3 jfmk-09-00145-f003:**
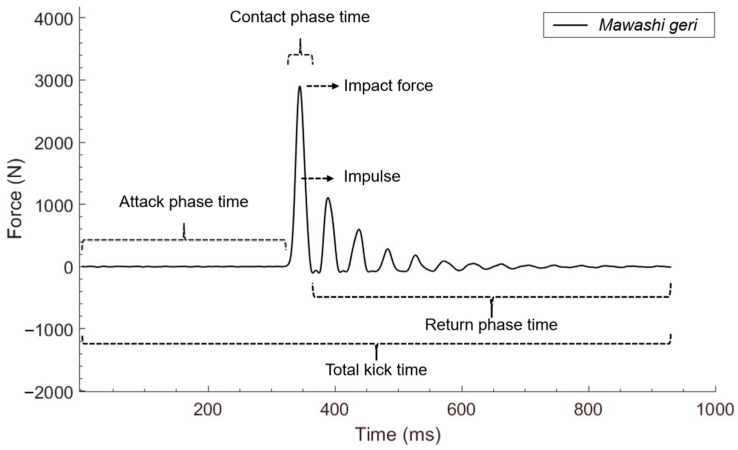
*Mawashi geri* variables. N: newtons; ms: milliseconds.

**Figure 4 jfmk-09-00145-f004:**
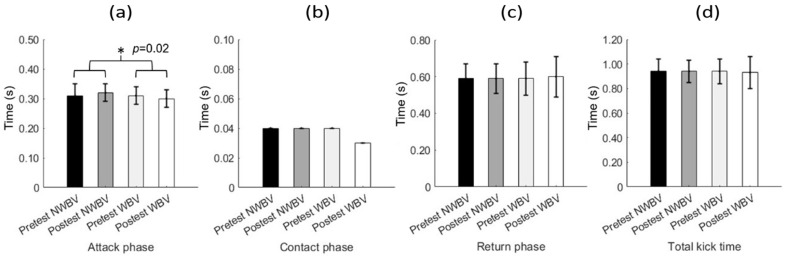
Kinematic data of *mawashi geri* in NWBV and WBV conditions and pre- and post-intervention moments. NWBV: No Whole-Body Vibration; WBV: Whole-Body Vibration; (**a**) attack phase time; (**b**) contact phase time; (**c**) return phase time; (**d**) total kick time. * Significant condition differences between NWBV and WBV.

**Table 1 jfmk-09-00145-t001:** Mean and standard deviation values of kinetic data for *mawashi geri* under NWBV and WBV conditions pre- and post-intervention.

Condition	NWBV		WBV		ANOVA		
Moment	Pre-Test	Post-Test	Pre-Test	Post-Test	Condition *p*	Moment *p*	Interaction *p*
Impact force (N)	2737 ± 985	2649 ± 894	2775 ± 1041	2662 ± 938	0.82	0.16	0.84
Impulse (N·s)	43.1 ± 11.18	42.1 ± 10.3	43.3 ± 12.0	42 ± 10.8	0.97	0.18	0.85

NWBV: No Whole-Body Vibration; WBV: Whole-Body Vibration; N: newtons; N.s: newton-seconds.

## Data Availability

The datasets analyzed during the current study are available from the corresponding author.

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
