# Peer review of "Whole-Body Vibration (WBV) as a Conditioning Activity for Roundhouse Kick (mawashi geri) Performance in Karate"

_jfmk, 2024, doi:10.3390/jfmk9030145_

Round 1

Reviewer 1 Report

Comments and Suggestions for Authors

Dear Authors

You have written an interesting paper focusing on the analysis of acute effects of WBV as a CA on the performance of the mawashi geri.

However, some parts need to be addressed for greater clarity and reproducibility of your study.

Introduction:

The introduction is well-written and leads nicely to the main study rationale.

Methods:

How did you determine your sample size (G*Power or any other method) - what is the power of your study? Report

What was their competitive level and belt degree - report.

The sample is very good height and body mass-wise; however, experience might impact your data—this is a limitation factor.

Experience 11.0 years  - training or competition - it is unclear - rewrite.

Any musculoskeletal injuries in recent years? report

In what part of the season were they? report

Were they in any weight loss protocol before competitions? report

Report how you performed the randomisation.

How many athletes were in each randomised group? report

Who performed skinfold measurements? Were they ISAK accredited? What was the itrarater reliability? report

Briefly describe the warm-up protocol for repeatability.

Why did you choose the 26 Hz and 4 mm amplitude? Based on which previous research? Elaborate and back up this decision.

Present a kinematic of the technique for greater understanding.

Who evaluated the kicks for correctness? report

Which result was taken into further analysis - the average or the best one? report

It looks like you are trying to input a new test that has not been used and validated before. This is not the best way! A huge limitation - Elaborate

The results show the possible problem with your sample from experience—the technique of competitors and beginners is completely different, and this is affecting your results.

Discussion:

You stated, ''Considering our findings, we can infer that WBV may be a valid method to activate PAPE before training.'' You didn't prove that, so rephrase and stick to the data you have proven. Don't assume, as it also might not be a good method according to your data.

Where is the limitation paragraph? Your study is far from perfect, especially regarding the sample size and the training-competition experience!

Conclusion - rephrase the last two sentences as you didn't study either of the elements you stated!!!

Kind regards

Comments on the Quality of English Language

Minor editing of the English language required

Reviewer 2 Report

Comments and Suggestions for Authors

The submitted manuscript aimed to investigate the acute effects of whole-body vibration (WBV) as a conditioning activity for roundhouse kick (mawashi geri) performance in karate. This study addresses an important and interesting topic. Additionally, the manuscript is well-written, with a clearly articulated rationale for the study and a well-described experimental design. However, there are significant concerns regarding the interpretation and discussion of the observed findings, as well as the conclusions drawn from this study. This reviewer believes that the manuscript requires substantial revisions to address the following issues:

 As reported in the Results section (L179-182 and L188-195), the partial eta-squared (η²) values for impact force condition, moment, and interaction were all 0.00. For impulse, only the moment showed a small effect with η² = 0.02, whereas the values for condition and interaction were 0. For the kinetic variables, five out of six variables showed partial η² values of 0, and the only other variable showed a small effect with η² = 0.02. Additionally, for the kinematic data of mawashi geri, no differences in contact or return phases or in total kick time between conditions WBV and NWBV were revealed. The only significant difference between conditions WBV and NWBV was for the attack phase analyzed by a two-way repeated measures ANOVA. However, this does not provide any information on whether the pre- vs. post-intervention values or the values compared at pre- or post-intervention between the two conditions differed for the attack phase. Despite this, under the Discussion section, the authors mentioned that “...athletes performed the attack phase in less time in the WBV condition” and that “...our findings revealed a statistically significant reduction in the attack phase time in the WBV condition” (L225-226). Furthermore, the authors wrote, “Considering our findings, we can infer that WBV may be a valid method to activate PAPE before training or competition...” (L244-246). According to this reviewer, the discussion reflects an exaggeration of the actual results observed in this study.

Lastly, under the Abstract (L24-27), the authors concluded, “... although WBV did not influence the kinetic variables of the mawashi geri in karate athletes, it shows potential for optimizing specific kinematic aspects of mawashi geri performance, particularly those related to attack speed.” The reviewer is not convinced that the observed results demonstrated a potential for optimizing specific kinematic aspects of mawashi geri performance, and such a conclusion seems not appropriate here.

Another major issue:

While the manuscript provides essential details about the vibration parameters and the equipment used, there are areas that require further elaboration. The manuscript does not describe the type of vibration (e.g., vertical or side-alternating) used in the study, nor does it mention the vibration magnitude. Additionally, there is no description of any procedure for calibrating the vibration platform (including the measurement location) to ensure that the specified vibration parameters were accurately delivered. What was the experimental room temperature? Furthermore, it is not clear whether the subjects were barefooted during exposure. By addressing these issues, the manuscript will improve in both transparency and reproducibility.

Comments on the Quality of English Language

Minor editing in English is required. A typical example: "Considering that in the study that evaluated WBV in a specific combat sports task, ... in combat sports." (L68-73)

Reviewer 3 Report

Comments and Suggestions for Authors
  • Lines 33-35: Mention that the definition of Kumite is specific to traditional karate forms. In other variations, such as Kyokushin, Shinkyokushin, Oyama, etc., Kumite refers to a fight without restrictions. As a karateka, this leads to a slight misunderstanding for me.

  • Lines 36-37: It is worth considering whether to include technical terms for kicks, such as mawashigeri jodan, in such cases.

  • Line 80: Specify exactly which style of karate the competitors represent, as karate currently encompasses a large number of styles and organizations. I know this information is provided at the beginning of the introduction, but it is crucial to include it in the description of the subjects. Was the sample size calculated?

The remaining methodology section is well-written, and the statistical analysis methods are appropriately chosen.

  • I suggest highlighting the limitations in a separate subsection in the discussion section.

  • Also, add an application conclusion.

Reviewer 4 Report

Comments and Suggestions for Authors

The title of the article is clear and precisely describes the main topic of the study, which is the effect of whole-body vibration on the performance of the mawashi geri technique in karate.

The article is generally well written, based on solid literature, the results are well presented and discussed in relation to the literature. The work is written in an understandable language, in accordance with the steps of the scientific method.

The introduction requires more depth in accordance with the importance of the work. It is worth expanding the introduction and discussion a bit. A more detailed explanation of the mechanisms through which vibration can affect performance is missing, which could strengthen the theoretical basis of the study.

The methodology is clear and detailed, but there is a lack of details regarding the criteria for the inclusion and exclusion of participants (e.g. in the table).

The results are presented clearly, and thanks to the use of tables and graphs visualizing the data, the reception is better.

The most important effect of the research, which shows that the reduction of the attack phase time under vibration conditions is interesting, however, as the authors themselves emphasize, the lack of interaction between conditions and moments of measurement requires caution in the interpretation of these results.

In the conclusions section, in my opinion, the conclusions should be more specific, maybe in points, and not general, but this is just a suggestion. It is worth adding, separating practical conclusions.

The final part of the publication mentions the need for further research, but potential directions for future research could be discussed in more detail.

Based on my description, it seems that the article can be improved, but it can also be published in its current form.

Round 2

Reviewer 1 Report

Comments and Suggestions for Authors

Dear Authors,

Thank you for adequately addressing all of my comments and suggestions. In my opinion, the quality of your manuscript has improved. Therefore, I recommend acceptance in its current form.

Kind regards

Comments on the Quality of English Language

Minor editing of English language required.